# Antibiotic Usage in Surgical Prophylaxis: A Retrospective Study in the Surgical Ward of a Governmental Hospital in Riyadh Region

**DOI:** 10.3390/healthcare10020387

**Published:** 2022-02-18

**Authors:** Nehad Ahmed, Mohamed Balaha, Abdul Haseeb, Amer Khan

**Affiliations:** 1Department of Clinical Pharmacy, Pharmacy College, Prince Sattam Bin Abdulaziz University, AL-Kharj 16273, Saudi Arabia; m.balaha@psau.edu.sa; 2Discipline of Clinical Pharmacy, School of Pharmaceutical Sciences, Universiti Sains Malaysia, Gelugor 11800, Malaysia; dramer@usm.my; 3Pharmacology Department, Faculty of Medicine, Tanta University, El-Gish Street, Tanta 31527, Egypt; 4Clinical Pharmacy Department, College of Pharmacy, Umm AlQura University, Makkah 21961, Saudi Arabia; amhaseeb@uqu.edu.sa

**Keywords:** antibiotic, appropriateness, prophylaxis, surgical, usage

## Abstract

Antibiotic prophylaxis is used to decrease the bacterial load in the wound to assist the natural host defenses in preventing the occurrence of surgical site infections. The present study aimed to investigate trends in using antibiotic prophylaxis in the surgical ward of a governmental hospital in the Riyadh Region and included collecting data concerning the use of antibiotic prophylaxis from medical electronic records. During 2020, most of the surgical patients received systemic antibiotics (82.40%). The most prescribed antibiotics were ceftriaxone (28.44%) and metronidazole (26.36%). The study also found that most of the patients received antibiotics for seven days or for five days, and only 1.08% of the patients received antibiotics appropriately for a maximum of one day. The present study showed that there was a major problem in selecting the correct antibiotic and in the duration of its use compared with the recommendations of the surgical prophylaxis guideline that was issued by the Saudi Ministry of Health. Thus, there is an urgent need to improve the adherence to the recommendations of surgical antibiotic prophylaxis guidelines in order to reduce the occurrence of negative consequences.

## 1. Introduction

Health care-associated infections (HCAIs) are infections that occur while receiving health care and that develop in health care facilities such as hospitals [1]. These infections first appear 48 h or more after patient admission or appear within 30 days after receiving health care [1]. HCAIs include catheter-associated urinary tract infections, ventilator-associated pneumonia, central line-associated bloodstream infections, and surgical site infections (SSIs) [2]. These infections are common and result in a high mortality rate. In the United States, there are 1.7 million reported HCAIs cases each year, causing about 100,000 deaths [3].

Surgical site infection (SSI) is a common hospital-acquired infection that causes significant health problems and results in prolonged hospitalization and increased treatment cost, in addition to increased patient mortality and morbidity [4]. SSIs are defined as infections that occur within 30 days of a surgery or within 90 days if the surgery includes the insertion of prosthetic material [5]. The World Health Organization reported that the pooled prevalence of surgical site infections was about 11.2 per 100 surgical patients [6]. It is estimated that about 10% of hospitalized patients in developing countries acquire HCAIs. Most of these infections are SSIs, which account for approximately 5.6% of surgically admitted patients [5].

Antibiotic prophylaxis is one of the measures used to decrease SSI incidence [7]. Antibiotic prophylaxis is used is to decrease the bacterial load in the wound to assist the natural host defenses in preventing the occurrence of an SSI [8]. Despite the effectiveness of prophylactic antimicrobials in preventing SSIs, the use of antibiotics is often incorrect [9]. Bedouch et al. stated that the inappropriate use of antibiotics occurs in 25% to 50% of general elective surgical procedures [10]. Thus, antibiotic prophylaxis is applied only if the costs and morbidity associated with infection are more than the costs and morbidity associated with antibiotic prophylaxis [11]. The unnecessary use of antibiotics and the use of broad-spectrum antibiotics increase the risk of resistance development [11]. Antimicrobial prophylaxis should be used for a short duration to reduce toxicity and antimicrobial resistance and decrease cost [12]. 

In Saudi Arabia, a few studies have been conducted, showing that antibiotic use patterns for surgical patients have changed over time, but there is still a problem in implementing antimicrobial stewardship practices. Ahmed et al. reported that surgeons in different Riyadh hospitals use preoperative antibiotic prophylaxis incorrectly [13]. Alghamdi et al. showed that although the Saudi Ministry of Health (MOH) devised a national antimicrobial stewardship plan to implement antimicrobial stewardship programs in Saudi hospitals, only 26% of hospitals reported the implementation of these programs [14]. Furthermore, Hammad et al. reported that in Aseer Hospital, the rate of adherence to preoperative and postoperative antibiotic prophylaxis guidelines was 36% and 56%, respectively, and that the average adherence rate was 46% [15]. Tolba et al. stated that there is a significant gap between current surgical antibiotic prophylaxis usage and international/national guidelines, and that there is a need for immediate action to ensure effective guideline adoption and implementation [16].

To improve the prescribing of antimicrobial prophylaxis, it is important to know the trends around prescribing antibiotics as well as the adherence to the recommendations of the guidelines. After that, appropriate antimicrobial stewardship practices should be implemented. The World Health Organization stated that the antimicrobial stewardship principles needing to be followed must give due consideration to the national and local context and the structure of the health system when carrying out antimicrobial stewardship activities [17]. In Saudi Arabia, the Ministry of Health devised a national antimicrobial stewardship plan that included a surgical prophylaxis guideline in 2018 to be implemented in governmental hospitals [18]. 

Although there are numerous studies on the use of antibiotics in general, there is a lack of studies on the use of antibiotic prophylaxis in our region. Therefore, the present study aimed to investigate trends in using antibiotic prophylaxis in the surgical ward of a governmental hospital in the Riyadh Region.

## 2. Materials and Methods

This was a retrospective study that included collecting data about the use of antibiotic prophylaxis from medical electronic records. The study was conducted in the surgical ward of a governmental hospital in the Riyadh Region. The hospital has an emergency ward, a maternity ward, and several other specialist wards that serve the public. 

This study included reviewing the medical records of patients who had a surgical procedure in the surgical ward during 2020. So, patients of both genders and from all age groups who visited the surgical ward were included, and patients in other departments were excluded. The total number of antibiotics included tablets, capsules, suspensions, vials, and ampules. Topical antibiotics such as drops and ointments were excluded from this study (Figure 1). 

The collected data included the total number of patients who had surgeries and the number and percentage of patients who received surgical antimicrobial prophylaxis. Furthermore, the collected data included the number of different antibiotics that were used and the age and gender of the patients receiving the antibiotics. Moreover, we collected data about the prescribed dosage forms of different antibiotics and the usage duration of these antibiotics. 

The data were collected using an Excel sheet and analyzed descriptively. The results were represented as numbers and percentages. The percentages were calculated by dividing each value by the total number and then multiplying the result by 100%. This study was approved by the ethical approval committee of the Saudi Ministry of Health with an IRB registration number H-01-R-053.

## 3. Results

### 3.1. Number and Percentage of Patients Who Received Antibiotics

During 2020, 915 patients had surgeries. Most of these patients received systemic antibiotics (82.40%) and less than 18% of these patients did not use antibiotics or topical antibiotics.

### 3.2. The Most Prescribed Antibiotics in the Surgical Ward

Table 1 shows the number of patients who used different antibiotics during this study. The most prescribed antibiotics were ceftriaxone (28.44%) and metronidazole (26.36%).

### 3.3. The Personal Data of the Patients Who Received Antibiotics in the Surgical Ward

Table 2 shows the personal data of the patients who received antibiotics. Most of the patients who received antibiotics were male (61.14%). Most of the patients were in the age groups of 30–39 years (22.81%), 20–29 years (20.29%), and 40–49 years (19.50%). 

Table 3 shows the prescribed dosage forms of antibiotics in the surgical ward. Most of the antibiotics were prescribed as a vial or ampule (90.69%), and 8.31% were prescribed as a capsule or tablet.

Table 4 shows the duration of the use of antibiotics in the surgical ward. More than 69% of the patients received antibiotics for seven days and 19.20% of them used antibiotics for five days. Only 1.08% of the patients received antibiotics for a maximum of one day.

## 4. Discussion

In this study, antibiotic prophylaxis was administered to 82.40% of cases. The rate of SSIs was very low (less than 0.5%) in the hospital, so almost all of the antibiotics were used as a prophylaxis and not for the treatment of SSIs. Several studies have shown that antibiotics are used excessively and incorrectly for the prevention of SSIs [19,20,21,22,23,24,25,26,27]. This study also showed that the most prescribed antibiotic was ceftriaxone, and that there is a high rate of using broad-spectrum antibiotics. The surgical prophylaxis guideline that was issued by the Ministry of Health and was implemented in governmental hospitals in Saudi Arabia recommended the use of first- or second-generation cephalosporins as a first line for most surgeries and not ceftriaxone [18]. Similarly to this result, Alemkere reported that ceftriaxone was used excessively and inappropriately in surgical prophylaxis, and that about 19.5% of the patients received a broad-spectrum antibiotic other than the antibiotics that are recommended by the guideline [28]. Similarly, Mohamoud et al. stated that nearly 84% of the surgical patients were given ceftriaxone, despite the drug not being mentioned in the national guideline [29]. Moreover, Van Kasteren et al. found that despite the availability of first-choice antibiotics, surgeons had been reported to fail to comply with the guideline recommendations [30]. They also reported that the barriers to the adherence to the guideline were a lack of awareness of appropriate guidelines, a lack of agreement of surgeons with the guideline recommendations, and logistic limitations in the surgical wards [30]. On the other hand, Oh et al. reported that the selection of antibiotics for 78.2% of surgical patients was consistent with the guideline recommendations [31]. Moreover, Al-Azzam et al. found that preoperative antibiotic prophylaxis was employed in almost all surgical departments of hospitals, and the choice of improper antimicrobials was ascribed to drug unavailability [32]. 

This study also found that most of the patients received antibiotics for seven days or for five days, and only 1.08% of the patients received antibiotics appropriately for a maximum of one day. Perioperative antibiotic prophylaxis should normally be discontinued within 24 h after surgery completion [33]. The Ministry of Health surgical prophylaxis guideline states that antibiotics should be used once, and if the surgery takes several hours, another dose of antibiotic could be given, but for a maximum of 24 h [18]. Similarly, Parulekar et al. reported that in a tertiary-care private hospital in India, the appropriateness of antibiotic selection was seen in 68%, and that the percentage of using the appropriate duration of antibiotics was 63% [34]. Musmar et al. found that in the Northwest Bank of Palestine, only 18.5% of surgical patients had appropriate antibiotic selection, and 31.8% of patients received antibiotics for an appropriate duration [35]. Moreover, Tourmousoglou et al. stated that for antibiotic prophylaxis in general surgery, the choice of antimicrobial agent was appropriate for about 70% of the patients and the duration of prophylaxis was optimal for about 36% [36]. Khan et al. reported that more than half (69%) of surgeons who participated in his study thought that antibiotics were overused in surgical procedures [37]. Furthermore, Oh et al. found that in the surgical ward at a tertiary hospital in Malaysia, prophylactic antibiotics were discontinued within 24 h after the operation in 77% of the cases [30]. Abdel-Aziz et al. reported that regarding antimicrobial prophylaxis in a tertiary general hospital, the overall use of antibiotics was 89%, but that the use of antibiotics did not match the recommended hospital protocols in more than 53% of cases [38]. They also reported that prolonged antibiotic use was the most common reason for nonadherence to antimicrobial prophylaxis guidelines (59.3%), followed by the use of an alternative antibiotic to that recommended in the protocol (31.5%) [38]. Gouvêa et al. conducted a review about the adherence to guidelines for surgical antibiotic prophylaxis and found that the rate of using the correct antibiotic choice ranged from 22% to 95%, and that the rate of the appropriate discontinuation of antibiotics ranged from 5.8% to 91.4% [39].

## 5. Limitation and Strength

The main limitation of the present study is that the rate of surgical site infections was not reported in the hospital, but the physicians informed that the rate of SSIs was less than 0.5%. This, the rate of surgical site infections (SSIs) might have been underestimated. Another limitation is that the diagnosis of the patients and the type of surgeries performed were not mentioned in the electronic files. A strength of this study is that we can estimate the appropriateness of using an antimicrobial agents before a surgery by comparing the commonly prescribed antibiotics and the duration of antibiotics used with the recommendations of the Saudi MOH guideline, because the recommended prophylactic antibiotics for the majority of surgeries were first-generation or second-generation cephalosporin antibiotics, and all of the prophylactic antibiotics should be used as a single dose or for a maximum of 24 h.

## 6. Conclusions

This study showed that antibiotics were administered to most of the surgical patients to prevent the occurrence of surgical site infections but that there was a major problem in selecting the correct antibiotic and in the duration of use compared with the recommendations of the surgical prophylaxis guideline issued by the Saudi Ministry of Health. There is an urgent need to improve the adherence to the recommendations of surgical antibiotic prophylaxis guidelines to reduce the occurrence of negative consequences. Moreover, it is important to encourage all healthcare providers to attend workshops and to be trained in the appropriate use of antibiotics for surgical patients.

## Figures and Tables

**Figure 1 healthcare-10-00387-f001:**
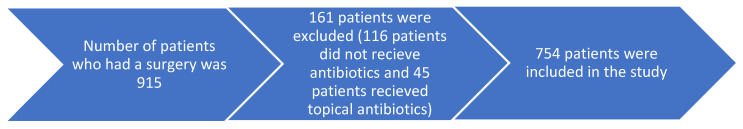
The number of patients included in this study.

**Table 1 healthcare-10-00387-t001:** Number of different antibiotics prescribed in the surgical ward.

Antibiotic	Number	Percentage
Ceftriaxone	397	28.44
Metronidazole	368	26.36
Piperacillin/Tazobactam	90	6.45
Cefazolin	71	5.09
Gentamicin	69	4.94
Amoxicillin/Clavulanic acid	68	4.87
Meropenem	67	4.80
Ciprofloxacin	59	4.23
Imipenem/Cilastatin	46	3.30
Cefuroxime	35	2.50
Linezolid	27	1.93
Vancomycin	26	1.86
Azithromycin	20	1.43
Amoxicillin	12	0.86
Ceftazidime	10	0.72
Clindamycin	10	0.72
Doxycycline	7	0.50
Cefotaxime	4	0.29
Tigecycline	4	0.29
Amikacin	1	0.07
Cefepime	1	0.07
Colistin	1	0.07
Clarithromycin	1	0.07
Moxifloxacin	1	0.07
Ceftazidime/Avibactam	1	0.07
Total number of antibiotics	1396	100

**Table 2 healthcare-10-00387-t002:** The personal data of the patients who received antibiotics in the surgical ward.

Variable	Category	Number	Percentage
Gender	Male	461	61.14%
	Female	293	38.86%
Age	<10	38	5.04%
	10–19	44	5.83%
	20–29	153	20.29%
	30–39	172	22.81%
	40–49	147	19.50%
	50–59	101	13.40%
	>59	99	13.13%

**Table 3 healthcare-10-00387-t003:** The prescribed dosage forms of antibiotics in the surgical ward.

Antibiotic	Capsule or Tablet	Syrup or Suspension	Vial or Ampule
Ceftriaxone	0	0	397
Metronidazole	35	1	332
Piperacillin/Tazobactam	0	0	90
Cefazolin	0	0	71
Gentamicin	0	0	69
Amoxicillin/Clavulanic acid	22	6	40
Meropenem	0	0	67
Ciprofloxacin	24	0	35
Imipenem/Cilastatin	0	0	46
Cefuroxime	1	0	34
Linezolid	0	0	27
Vancomycin	0	0	26
Azithromycin	17	3	0
Amoxicillin	8	4	0
Ceftazidime	0	0	10
Clindamycin	0	0	10
Doxycycline	7	0	0
Cefotaxime	0	0	4
Tigecycline	0	0	4
Amikacin	0	0	1
Cefepime	0	0	1
Colistin	0	0	1
Clarithromycin	1	0	0
Moxifloxacin	1	0	0
Ceftazidime/Avibactam	0	0	1
Total	116 (8.31%)	14 (1.00%)	1266 (90.69%)

**Table 4 healthcare-10-00387-t004:** The duration of the use of antibiotics in the surgical ward.

Antibiotic	1 Day	2 Days	3 Days	4 Days	5 Days	6 Days	7 Days
Ceftriaxone	3	14	25	3	57	0	295
Metronidazole	0	1	10	4	104	1	248
Piperacillin/Tazobactam	0	0	1	0	8	0	81
Cefazolin	6	3	27	0	21	1	13
Gentamicin	1	1	22	0	18	0	27
Amoxicillin/Clavulanic acid	3	1	6	1	10	1	46
Meropenem	0	0	0	0	0	3	64
Ciprofloxacin	0	0	3	0	12	0	44
Imipenem/Cilastatin	0	0	0	0	2	0	44
Cefuroxime	0	0	1	0	12	0	22
Linezolid	0	0	1	0	4	0	22
Vancomycin	0	0	1	0	1	0	24
Azithromycin	0	1	7	0	6	2	4
Amoxicillin	0	0	0	0	5	0	7
Ceftazidime	0	0	0	0	2	0	8
Clindamycin	0	0	2	0	4	0	4
Doxycycline	0	0	0	0	0	0	7
Cefotaxime	1	0	0	0	2	0	1
Tigecycline	1	0	0	0	0	0	3
Amikacin	0	0	0	0	0	0	1
Cefepime	0	0	0	0	0	0	1
Colistin	0	0	0	0	0	0	1
Clarithromycin	0	0	0	0	0	0	1
Moxifloxacin	0	0	0	0	0	0	1
Ceftazidime/Avibactam	0	0	0	0	0	0	1
Total	15 (1.08%)	21 (1.51%)	106 (7.59%)	8 (0.57%)	268 (19.20%)	8 (0.57%)	970 (69.48%)

## Data Availability

Not applicable.

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
