# Peer review of "Antibiotic Usage in Surgical Prophylaxis: A Retrospective Study in the Surgical Ward of a Governmental Hospital in Riyadh Region"

_healthcare, 2022, doi:10.3390/healthcare10020387_

Round 1

Reviewer 1 Report

I enjoyed reading Nehad et all manuscript on Antibiotic Usage in Surgical Prophylaxis. All data that could reduce antibiotic misusing or overusing are welcome and should be of interest to scientists in the antimicrobial resistance field.

The authors might consider the following comment to improve the manuscript.

  1. Line 34 “In the United States, HCAIs reported in 1.7 million cases each….” please check something is missed to make the sentence understandable.
  2. Line 44: Kefale et al (Reference 4) study was performed one setting study (Ethiopia) not many LMICs, please delete or change the formulation of this statement.
  3. Despite the authors reporting a lack of studies about using antibiotic prophylaxis in their region, what is missed is the available study performed in other hospitals in the region. Please report the data available (lack is different from absence)
  4. Most important: Please provide information on guidelines regarding antibiotic prophylaxis in your region /Hospital.

Method

  1. Please provided a flowchart showing the total of patients medical assessed, those excluded, (with the reason), included patient

Result

  1. I would suggest to the author describe the characteristic (demographic, surgical diagnosis category, etc) of included patients
  2. How many eligible patients were hospitalized in the surgical ward during the study period? This should be included in the main text.
  3. Please for each proportion use xx/xx(%) or any other style you like, but the reader should have the numerator and the denominator of your proportion.
  4. If my above comment is considered, table 1 could be deleted. add key information in the table in the main text and delete it.
  5. Table 3: Please, what is the rationale behind splitting the data according to gender? If there is no strong evidence of an association between sex and antibiotic used, I would suggest you delete it

Discussion

  1. Line 116-117-Please add that Alemkere study setting should be applied for all papers cited in the discussion. This is very important to improve the understanding of the reader and to make sure that the studies are comparable.

Limitation:

  1. Were you able to see all of the information you were looking for in the patient medical file? If not add that as a limitation.
  2. Why does the study focus only on one year? Any reason behind that could be added as a limitation.
  3. Who generalizable is your study to the region or other hospital in the country?

Strength:

  1. If you think there is some strength to your study please add.

Conclusion:

  1. “The study showed that antibiotic was administered for most of the surgical patients to prevent the occurrence of surgical site infections but the there was a major problem in selecting the correct antibiotic and in the duration of using it”.
  2. Your study did not support this conclusion. The result did not report: Problem in selecting antibiotic nor the duration?  
  3. Did the hospital/region have a guideline that supports your conclusion? Please provide the detail to make sure the reader will understand your paper.

Author Response

We modified the sentence in line 34 and deleted the sentence in line 44.

We added data about the conducted studies and the implemented guideline.

We added a flowchart for inclusion and exclusion criteria.

We make a new table for personal data and deleted the 2 tables for age and gender. moreover, the tables now contained numbers and percentages. 

we rearranged some sentences in the discussion. now the first paragraph was for the appropriateness of antibiotic selection and the other paragraph was about the duration of antibiotic use. We also make a comparison with the ministry of health guideline that was implemented in the governmental hospitals. 

We added the limitations and strengths of the study.

we add to the conclusion that the inappropriateness in the study was by selecting broad-spectrum antibiotics for a long duration as compared to MOH guidelines. 

All of the modifications were written in the red color 

Reviewer 2 Report

In the introduction, cite past studies on antibiotic usage in Saudia Arabia and other Middle East countries as these are relevant context to local prescribing patterns. Discuss how how these patterns have changed over time - or not.

Also introduce the concept that antimicrobial stewardship principles need to be followed, and cite the relevant WHO document https://www.who.int/publications/i/item/9789240025530

In the results, show an analysis of the type of surgery and the pattern of antibiotic use. This is important for understanding the mix of surgical procedures at the hospital and the procedures for which no antibiotic prophylaxis was used. Explain if there are local rules regarding prescribing practices (rule-based prescribing).

For Table 3, include percentages, and analyze whether there were prescribing differences for male versus female patients.

For Tables 4 and 6, rearrange from most commonly prescribed to least, as this is more logical and matches the sequence in Tables 2 and 3.

Table 5 is not necessary, and instead the text can include the summary totals from the bottom of the table.

Line 111 - Where did the SSI statistic come from? How does this relate to line 155?

The discussion needs to compare data from this study for Riyadh to similar studies in other locations - and summarise the data (e.g. use a table, and perform analyses of frequency).

The discussion does not have a good logical flow, and is written in a passive voice. It does not clearly state what the overall message from the study is.
Use one idea per paragraph. Devote time to discussing the issues with prescribing versus WHO and other guidelines on antimicrobial stewardship.

Overall, the paper does not add much to current literature, and this is especially so given the lack of any statistical comparisons to other studies (from the region or elsewhere), or within the study data itself.

The paper has many small spelling and style errors in every section of the text and needs professional English language editing.  There are also line spacing errors that need to be fixed.

e.g. Line 26 infection - should be - infections
Line 40  World Health Organization reported that the pooled prevalence of overall Surgical site infections - should be -  The World Health Organization reported that 40
the pooled prevalence of surgical site infections
Line 42 patients in the developing  - should be - patients in developing 

Line 44 kefale et al reported that more than one in ten surgical patients in low- and middle-income countries developed SSI  - this repeats information in line 42

Table 2 has a formatting / alignment error (Gentamicin)

The references have style errors (incorrect capitalization in the titles of some articles, e.g. references 4, 6, 9, 11, 23, and 32.

Author Response

We add the studies that were conducted in Saudi Arabia regarding surgical prophylaxis.

We add the principles of implementing antimicrobial stewardship as mentioned by WHO.

In the results, we add a new table about personal data and deleted the 2 tables about gender and age.

We add information about the ministry of health guideline that was implemented in governmental hospitals.

We add a limitation that the type of surgery was not mentioned in the electronic files. 

The tables now contained numbers and percentages. 

The antibiotics were rearranged from the most commonly used to the least.

We modified some grammatical errors.

The sentences in the discussion were rearranged to two main paragraphs one about the appropriateness of antibiotic selection and the other about the appropriateness of antibiotic duration.

We also add a comparison between the practice and the recommendations of MOH guideline that was implemented in the hospital. 

The sentences in lines 26,40, 42 were modified.

The sentence in line 44 "kefale et al reported" was deleted. 

The formatting/alignment error (Gentamicin) was modified.

The references that have style errors were modified.

The corrections were written in red color

Round 2

Reviewer 1 Report

Dear authors thank you for providing a response to the comments.

However, I am not able to follow the author's response to my comments.

Please respond point by point to my comments.

Author Response

  1. Line 34 “In the United States, HCAIs reported in 1.7 million cases each….” please check something is missed to make the sentence understandable.

I rewrite the sentence

  1. Line 44: Kefale et al (Reference 4) study was performed one setting study (Ethiopia) not many LMICs, please delete or change the formulation of this statement.

I deleted this statement

  1. Despite the authors reporting a lack of studies about using antibiotic prophylaxis in their region, what is missed is the available study performed in other hospitals in the region. Please report the data available (lack is different from absence)

I mentioned in red color the results of the available studies

  1. Most important: Please provide information on guidelines regarding antibiotic prophylaxis in your region /Hospital.

I add information about the implemented guideline which is the ministry of health guideline

Method

  1. Please provided a flowchart showing the total of patients medical assessed, those excluded, (with the reason), included patient

I added a flowchart about the inclusion and exclusion criteria

Result

  1. I would suggest to the author describe the characteristic (demographic, surgical diagnosis category, etc) of included patients

I add a table about the personal data of the patients

  1. How many eligible patients were hospitalized in the surgical ward during the study period? This should be included in the main text.

I add a flowchart for the eligible patients and the patients who were excluded

  1. Please for each proportion use xx/xx(%) or any other style you like, but the reader should have the numerator and the denominator of your proportion.

I represented the data as numbers and percentages so the number and percentage of each variable were written in the table  

  1. If my above comment is considered, table 1 could be deleted. add key information in the table in the main text and delete it.

I deleted table 1 and wrote the number of patients who had surgery and included in the study and the patient who were excluded from the study in the main text and in the flowchart

  1. Table 3: Please, what is the rationale behind splitting the data according to gender? If there is no strong evidence of an association between sex and antibiotic used, I would suggest you delete it

I deleted the 2 tables about sex and age as requested by one reviewer and include the data about age and gender in a table about the personal data of the patients.

Discussion

  1. Line 116-117-Please add that Alemkere study setting should be applied for all papers cited in the discussion. This is very important to improve the understanding of the reader and to make sure that the studies are comparable.

I rearranged the sentences and made 2 main paragraphs: one about the appropriateness of antibiotics selection and the other about the appropriateness of antibiotic duration. Moreover, I add one sentence in each paragraph about the recommendations of the ministry of health guideline that were implemented in the hospital.

Limitation:

  1. Were you able to see all of the information you were looking for in the patient medical file? If not add that as a limitation.
  2. Why does the study focus only on one year? Any reason behind that could be added as a limitation.
  3. Who generalizable is your study to the region or other hospital in the country?

I added 2 limitations, particularly the unavailability of the type of surgery that the patient had in the hospital

Strength:

  1. If you think there is some strength to your study please add.

I add one strength “although the type of surgeries was not found in the files but we can estimate the appropriateness of the antibiotic because the guideline that was implemented in the hospital and the international guidelines recommended the use of first or second generation cephalosporin but in the study the most commonly used antibiotics was ceftriaxone which are broad spectrum third generation cephalosporin. Moreover, the hospital guideline and the international guidelines recommended the use of single dose antibiotic and if more doses are needed the surgeons prescribe surgical prophylaxis for 24 hours only.

Conclusion:

  1. “The study showed that antibiotic was administered for most of the surgical patients to prevent the occurrence of surgical site infections but the there was a major problem in selecting the correct antibiotic and in the duration of using it”.
  2. Your study did not support this conclusion. The result did not report: Problem in selecting antibiotic nor the duration?  
  3. Did the hospital/region have a guideline that supports your conclusion? Please provide the detail to make sure the reader will understand your paper.

I add data about the guideline recommendations and the inappropriateness because the hospital and the international guidelines recommended the use of first- or second-generation cephalosporin but in the study the most commonly used antibiotics was ceftriaxone which are broad spectrum third generation cephalosporin. Moreover, the hospital guideline and the international guidelines recommended the use of single dose antibiotic and if more doses are needed the surgeons prescribe surgical prophylaxis for 24 hours only.

Reviewer 2 Report

The changes made have addressed my earlier concerns and improved the overall quality of the paper.

Author Response

Thanks